# High-Performance Photodetectors Based on the 2D SiAs/SnS_2_ Heterojunction

**DOI:** 10.3390/nano12030371

**Published:** 2022-01-24

**Authors:** Yinchang Sun, Liming Xie, Zhao Ma, Ziyue Qian, Junyi Liao, Sabir Hussain, Hongjun Liu, Hailong Qiu, Juanxia Wu, Zhanggui Hu

**Affiliations:** 1Tianjin Key Laboratory of Functional Crystal Materials, Institute of Functional Crystal, Tianjin University of Technology, Tianjin 300384, China; acencore@163.com (Y.S.); 18837161430@163.com (Z.M.); hjliu@email.tjut.edu.cn (H.L.); 2CAS Key Laboratory of Standardization and Measurement for Nanotechnology, CAS Center for Excellence in Nanoscience, National Center for Nanoscience and Technology, Beijing 100190, China; xielm@nanoctr.cn (L.X.); qianzy2019@nanoctr.cn (Z.Q.); liaojy2018@nanoctr.cn (J.L.); sabirphys@yahoo.com (S.H.); 3University of Chinese Academy of Sciences, Beijing 100049, China

**Keywords:** SiAs/SnS_2_, heterojunction, photodetectors

## Abstract

Constructing 2D heterojunctions with high performance is the critical solution for the optoelectronic applications of 2D materials. This work reports on the studies on the preparation of high-quality van der Waals SiAs single crystals and high-performance photodetectors based on the 2D SiAs/SnS_2_ heterojunction. The crystals are grown using the chemical vapor transport (CVT) method and then the bulk crystals are exfoliated to a few layers. Raman spectroscopic characterization shows that the low wavenumber peaks from interlayer vibrations shift significantly along with SiAs’ thickness. In addition, when van der Waals heterojunctions of *p*-type SiAs/*n*-type SnS_2_ are fabricated, under the source-drain voltage of −1 V–1 V, they exhibit prominent rectification characteristics, and the ratio of forwarding conduction current to reverse shutdown current is close to 10^2^, showing a muted response of 1 A/W under excitation light of 550 nm. The light responsivity and external quantum efficiency are increased by 100 times those of SiAs photodetectors. Our experimental results enrich the research on the IVA–VA group *p*-type layered semiconductors.

## 1. Introduction

Benefitting from novel optical and photoelectric properties, the application of two-dimensional layered materials in the field of photodetection has attracted widespread attention [1,2,3,4,5]. Group IVA–VA (IVA = Si, Ge, VA = P, As) layered materials are an essential branch of two-dimensional (2D) materials. In the past years, researchers have mainly focused on calculating the energy band structure of such materials and inferred that the single-layer structure has a broader bandgap (about 2.5 eV), which is significantly larger than the macroscopic bulk material [6,7]. Compared with common two-dimensional materials such as MoS_2_ and WS_2_, the wider forbidden bandwidth and energy band position is beneficial to the absorption of visible light photons and their application in the field of photocatalysis [8,9]. Due to their inherent high in-plane anisotropy, such materials are significant in designing and applying photodetection, polarization sensor devices, and angle-dependent electronics [10,11,12,13,14,15,16,17]. However, due to the limitation of material synthesis methods (compounds containing P and As elements are challenging to synthesize in the atmospheric environment controllably), the layer-dependent effect and device optoelectronic properties of this material require more research. The research on the 2D properties of IVA–VA materials has been recently supplemented and improved. 2D GeAs has excellent thermoelectric properties along the b-axis (growth direction) [18]. Researchers have also confirmed the high-temperature stability of 2D SiP [19] Polarized Raman spectroscopy of 2D GeP, and angle-dependent electronics are investigated in more detail [20]. However, there are still few studies on the specific properties of 2D SiAs crystals [21,22,23].

For the SiAs crystals, the research has mainly focused on calculating the energy band structure, synthesis, and structure [23,24,25]. These studies on SiAs show their broad application prospects in the field of nanoelectronics. However, experimental research on the optical and optoelectronic properties is still lacking. Recently, Jeunghee Park et al. achieved photodetectors with SiAs nanosheets that exhibit high performance in the UV-visible region [26]. However, because of the abundant defects and many-body substantial effects, the performance of photodetectors of 2D SiAs is deficient compared to transition metal dichalcogenides (TMDCs) materials, which restricts its further applications. Therefore, developing methods to improve the performance photodetector of 2D SiAs is an important issue that should be addressed. Previous studies have indicated that chemical treatment, oxygen bond doping, and electrostatic doping can improve the performance of photodetectors of 2D materials [27,28,29]. However, these methods are not suitable for SiAs due to their weak chemical stability. The construction of heterostructures is proven to be an efficient way to engineer the physical properties while maintaining the intrinsic nature of each component. The graphene–WS_2_–Si (*n*-type) van der Waals heterostructure, with the WS_2_ layer inserted between graphene and the Si, results in the apparent rectification behavior, a broad spectrum response (from visible light to near-infrared) as the heterojunction photodetector, and excellent photodetection performance (maximum photoresponsivity of 54.5 A/W at 800 nm) [30]. In addition, from the perspective of changing the contact characteristics of the electrodes, a high-performance electronically complementary multilayer WS_2_ asymmetric Ni and Gr/Ni hybrid contact device is constructed. The novel device structure converts the carrier type in WS_2_ from *n*-type to *p*-type, with a current rectification exceeding 10^4^, a maximum optical response rate of 4 × 10^4^ A/W at a wavelength of 532 nm, and high-performance photovoltaic capability [31]. Therefore, developing heterostructures with specific structures would be an efficient way to realize devices with high-performance photodetectors of 2D SiAs.

Herein, high-quality large-size SiAs single crystals have been successfully obtained by the chemical vapor transport (CVT) method. In order to improve the photoelectric detection performance of SiAs, van der Waals *p–n* junctions have been constructed using SnS_2_ as an *n*-type contact material. Compared with pure SiAs, *p–n* junctions show higher sensitivity in the visible range, including the SiAs/SnS_2_ heterojunction exhibiting a 1 A/W responsivity under 550 nm laser irradiation, and the corresponding photoconductive gain or external quantum efficiency (EQE) is 1.0 × 10^5^. The light responsivity and external quantum efficiency are increased by 100 times those of SiAs photodetectors. This research further supplements the gap in the research of IVA–VA semiconductors. The preparation of SiAs crystals by the CVT method provides a reference for the growth of IVA–VA group crystals and the high-quality synthesis of other new crystal materials. In addition, our work shows that *p*–SiAs crystals are a kind of optoelectronic material with potential research value and provide experience for the further construction of functional devices, such as *p–n* junctions of the IVA–VA series layered semiconductors. Finally, it expresses the broad application prospects of this type of material in the field of nanoelectronics in the future.

## 2. Experimental Section

### 2.1. Synthesis of SiAs Crystal

The CVT reaction system includes reactants and a transport agent that transports the gaseous reactants under a temperature gradient [32]. In this experiment, high-purity arsenic blocks (99.9%, Hawk, Beijing, China) and silicon powder (99.9%, Alfa, Shanghai, China) were used as raw materials, and iodine crystals (99.9%, Alfa) were used as transport agents for the CVT reaction. The mixture of As, Si, and I_2_ (0.51 g) with the mole ratio As:Si:I_2_ = 1.01:1:0.025 was placed in a vacuum-sealed quartz tube (length 15 cm, inner diameter 1.8 cm, wall thickness 1 mm) and quickly heated to 500 °C, the temperature slowly raised to 1050 °C, and maintained for 100 h, then decreased to 500 °C at a rate of 0.08 °C/min, and then quickly dropped to room temperature. SiAs strips crystals were synthesized with a bright luster. Most of the SiAs crystals were 2 cm × 0.15 cm (some were up to 3–4 cm in length).

### 2.2. Characterization

SiAs few-layer samples were prepared by mechanical exfoliation of single crystals onto a 300 nm SiO_2_/Si substrate using Scotch tape, then distinguished by optical microscopy (OM, on an Olympus BX51 microscope). AFM (Bruker Corp., Billerica, MA, USA, Dimension Icon) imaging was carried out in the Ar-filled glovebox (Mikrouna, Shanghai, China, Super 1220/750, H_2_O < 0.1 ppm, O_2_ < 0.1 ppm) by using an insulating silicon AFM tip (Bruker Corp., k = 26 N m^−1^, f0 = 300 kHz) in the mode of PeakForce QNM (Quantitative Nano Mechanics). Raman spectra were carried out on a home-built vacuum, variable temperature, low-wavenumber Raman system with 532 nm excitation. A NA = 0.82 low-temperature objective (LT-APO/VIS/0.82, attocube systems AG, Munich, Germany) was used for laser focusing and signal collection. The laser power was kept below 1 mW μm^−2^ to avoid damage to the sample. Amplified-spontaneous emission (ASE) filters, a beam splitter, and notch filters (Ondax Inc., Monrovia, CA, USA) were used to achieve low wavenumber detection down to 10 cm^−1^. The intensity and peak position of Raman modes were fitted using the Lorentz functions.

### 2.3. Transfer Methods

PDMS films used in the transfer process were prepared using SYLGARD 184 (Dow Corning Corporation, Midland, MI, USA), a two-part kit consisting of prepolymer (base) and cross-linker (curing agent). We mixed the prepolymer and cross-linker at a 10:1 weight ratio and cured the cast PDMS films on SiO_2_/Si wafers at 100 °C for 4 h. During the transfer process, the PDMS/SiAs films were clamped by a manipulator equipped with homemade step-motor linear guides to assist their peeling-off from long strips of source material and stamping onto receiving substrates. A similar method was used to obtain SnS_2_ flakes by mechanical peeling on a transparent PDMS film. With the help of an optical microscope, SnS_2_ and SiAs were aligned and transferred one by one. These devices are produced by dry transfer technology under Ar-filled conditions. After transfer, all samples were annealed at 200 °C for 2 h, under the protection of 20 sccm H_2_/140 sccm Ar gas, ~1 Torr.

### 2.4. Device Fabrication and Measurements

The artificially constructed heterojunction device is constructed by standard electron beam lithography (EBL, FEI Quanta 650 SEM, and Raith Elphy Plus). Cr/Au (10 nm/70 nm) were deposited as contact electrodes using thermal evaporation. A semiconductor parameter analyzer (Keithley, Bradford, UK, 4200-SCS) and standard probe station were used for electronic and photoelectric measurement of the device (the spot diameter of the 550-nm laser is 2 mm), using adjustable power and an incident wavelength laser to measure the optical response of the device.

## 3. Results and Discussion

### 3.1. Synthesis of SiAs Crystal

The experimental configurations used in our growth process are shown in Figure 1a. Bulk SiAs single crystals were fabricated by the CVT method. The SiAs nanosheets were prepared by mechanical exfoliation with Nitto tape of a SiAs single crystal, as shown in Figure 1a. In a typical CVT run, the precursor powder’s sealed ampoule is horizontally loaded into a furnace with a high-temperature source zone and a low-temperature growth zone (Figure 1b). The furnace is heated to 1050 °C at a heating rate of 20 °C/min, maintained for one week, and naturally cooled to room temperature. After the reaction, the ampoule is broken and the sample is removed from the ampoule.

Similar to GeAs [18,33], SiAs is also a layered semiconductor crystal, which belongs to the space group C2/m (No. 12), and the lattice parameters are a = 15.949 Å, b = 3.668 Å, c = 9.529 Å, and β = 106° [26]. The Si–Si dumbbell is surrounded by a twisted triangular antiprism formed by three arsenic (As) atoms in each layer. In the two directions of formation, one is almost parallel to the layer, and the other is perpendicular to the layer, as shown in Figure 1c, where the green balls are As atoms and the blue balls are Si atoms. SiAs crystal is covalently bonded in the inner layer and terminated by As atoms in each layer, while the interlayer is stacked together with van der Waals interaction.

### 3.2. Basic Characterization of SiAs Crystal

The preliminary characterization information of the SiAs crystals is shown in Figure 2a. Although the XRD pattern is highly overlapped with the standard card, after comparison, the lattice parameter information is consistent with the theoretical calculation parameters, indicating the high purity and few miscellaneous items of our growth SiAs crystals. In addition, SiAs crystals show a strong ($\overset{-}{2}01$) peak, suggesting that SiAs crystals grow along the ($\overset{-}{2}01$) plane (Appendix A).

Figure 2b shows the Raman spectrum of SiAs crystal. Several Raman peaks in the range of 0~600 cm^–1^ can be collected, and the assignment for each peak is also labeled according to the analysis of molecular vibration mode in SiAs crystal [34]. Among them, phonon modes with A_g_ symmetry are dominant. For example, phonon modes with A_g_ symmetry have moderate intensity peaks at 91 cm^–1^ and 114 cm^–1^ in the 80–140 cm^–1^ region. There are high-intensity peaks at 165 cm^–1^ (A_g_) and 189 cm^–1^ (A_g_). Around 400 wavenumbers, 371 cm^–1^ (B_g_), 390 cm^–1^ (A_g_), and 419 cm^–1^ (A_g_) have medium intensity peaks. In addition, around 520 cm^–1^ shows a moderate intensity peak of 518 cm^–1^ (A_g_). After the actual measurement results of statistical Raman analysis, except for the fragile phonon mode with A_g_ at 109 cm^–1^, there are a total of 17 phonon modes (among them, there are 12 phonon modes with A_g_ symmetry and five phonon modes with B_g_ symmetry). This is consistent with the number and pattern of theoretical calculations [34]. X-ray photoelectron spectroscopy (XPS) can indicate the elemental composition of the synthesized product from the perspective of an atomic orbital. As shown in Figure 2c,d, the Si 2*p* orbit of SiAs as grown by the CVT method are located at 103.4 eV and 100.2 eV, which are suitable for Si 2*p*_3/2_ and Si 2*p*_1/2_, and the binding energies of As 3*d*_3/2_ and As 3*d*_5/2_ are 45.4 eV and 41.8 eV, respectively, which is consistent with the information of SiAs crystals. XPS data of SiAs crystal exclude the presence of SiAs_2_ and AsI_3_ and other impurities in the synthesized product, further proving the high purity of our growth SiAs.

Regarding the bandgap discussion of strip SiAs crystals, we performed a U-V ultraviolet diffuse reflection on powdered SiAs after grinding, as shown in Appendix A. The fitted value of the spectrum was close to 1.45 eV, which is roughly consistent with the previous report [6,7,26]. In addition, according to the theoretical calculation of PBE, the forbidden bandwidth of monolayer SiAs was close to 1.7 eV [6,7,9].

### 3.3. Atomic-Level Morphology Characterization of SiAs Crystal

The typical STEM image in Figure 2e shows clear lattice fringes, and the crystal face index is (010). The Fast Fourier Transform (FFT) shows the properties of single crystals (low-magnification topographic map, reference Appendix A). The measured (001) plane spacing is 6.1 Å, and (110) plane spacing is 3.6 Å, which corresponds well to the structural information of SiAs. The FFT in the illustration also clearly shows the diffraction points of (200) and (001). Energy Dispersive X-ray Spectroscopy (EDX) can confirm the uniform distribution of silicon atoms and arsenic atoms, and the atomic ratio is close to 1:1. Please refer to the Appendix A to screen the individual Si atom distribution and As atom distribution. The above basic characterization is sound proof that we have synthesized high-quality SiAs crystals and warrants more in-depth structural analysis and performance research.

### 3.4. Low Wavenumber Raman Vibration Mode of SiAs Crystal

For 2D materials, Raman spectroscopy can be used to characterize the structure of 2D materials (layer number, lattice orientation, etc.) through the peak position, intensity, and full width at half maxima (FWHM) of Raman modes [35,36,37].

Here, SiAs samples with different layers were obtained by mechanical exfoliation. Raman spectra of few-layer SiAs samples with different thicknesses are shown in Figure 3a. The atomic force microscope (AFM) images were shown in Figure 3b–h, and the thickness was labeled. Considering a layer-to-layer spacing of 0.7 nm [23,24], we identified the few-layer SiAs with thickness down to about 1.16 nm, corresponding to a two-atomic layer of SiAs. It is worth noting that the Raman spectra of SiAs samples for thickness below 3.3 nm cannot be obtained due to the small size of about 1~2 μm for these thin samples (Figure 3c) and the reduced contrast in the vacuum chamber. Nevertheless, several Raman peaks can be detected on all the SiAs samples, and the Raman modes were assigned according to the previous works. We fitted the Raman peaks with the Lorentz function and found that most Raman modes at the range of 100~600 cm^–1^ do not show a noticeable shift as the sample thickness increases. Only a few Raman modes slightly shift to the higher frequency, i.e., A_1g_ modes lying at ~112 cm^–1^ and 165 cm^–1^, as shown in Figure 3i,j.

The low-frequency rigid vibrational modes were explored for few-layer SiAs, which are the relative vibrations of individual SiAs layers perpendicular or parallel to the layer plane and are usually located in low frequency below 100 cm^–1^ due to the relatively weak interlayer interaction, similar to other 2D materials [38,39,40]. Fortunately, two Raman peaks were observed in the range of 10~40 cm^–1^. Unlike the high-frequency Raman modes, these two modes showed a significant redshift as the sample thickness increased (as shown in Figure 4a,b). The relationship between the frequencies of rigid-layer modes and layer number can be analyzed by establishing a linear chain model, where the nearest-neighbor interlayer coupling is considered [38,39,40]. The layer-dependent frequencies of layer breathing and shear modes for the same branch can be given by
(1)$$\omega (N)=\frac{1}{\sqrt{2}\pi c}\sqrt{\frac{\alpha }{\mu }}\sqrt{1\pm \cos(\frac{\pi }{N})}$$
where *N* is the number of SiAs layers, *μ* = 2.61 × 10^−6^ kg/m^2^ is the mass per unit area for monolayer SiAs, α is the strength of the interlayer coupling, c is the speed of light in cm/s, and minus and plus signs correspond to the breathing and shear modes, respectively. In the case of bilayer SiAs,
(2)$$\omega (2)=(1/\sqrt{2}\pi c)\sqrt{\alpha /\mu )}$$

Therefore, the above expression becomes
(3)$$\omega (N)=\omega (2)\sqrt{1\pm \cos(\pi /N)}$$

By considering thickness-dependent low-frequency peak position and the linear-chain model, the observed two Raman peaks can be assigned to the layer breathing modes. Moreover, the results can be well fitted with the linear chain model, and thus we can derive the out-of-plane force constant *k_z_* ~ 6.98 × 10^19^ N/m^3^.

### 3.5. Photoelectric Correspondence of SiAs/SnS_2_ Heterojunction

It is reported that IVA–VA group layered semiconductor materials exhibit unique *p*-type semiconductor characteristics [19,20,41,42,43]. We used mechanical peeling and transferring to construct *p–n* junction devices containing a few layers of *p*-type SiAs. Relatively stable few-layer SnS_2_ *n*-type materials were chosen to explore the electrical and optoelectronic properties of the heterojunction under irradiated light (550 nm).

The operating principle of the *p*–SiAs/*n*–SnS_2_ van der Waals heterojunction photodetector can be understood from the schematic diagram of the device and band diagram of the heterostructure shown in Figure 4. Figure 5a shows that the thinner SnS_2_ is stacked on SiAs with bottom gate SiO_2_ by mechanical peeling and dry transfer. Relatively thick SiAs crystals were chosen due to concerns about the effect of SiAs crystal stability on the experimental results. The inset of Figure 5a,i shows the optical topography image of the device, and (ii) is the AFM image of the heterojunction region. It can be seen from the AFM picture that the thickness of the upper layer of SnS_2_ is 7.5 nm, and the thickness of the lower layer of SiAs is close to 100 nm. The crystal quality is characterized by Raman spectroscopy; Figure 5b is the Raman spectroscopy of different positions (marked with colored dots) in the topography of the device in the inset (i) of Figure 4a. The obtained Raman spectroscopy, according to the black circle mark, shows three medium intensity peaks corresponding to the SiAs sample at 371 cm^–1^ (B_g_), 390 cm^–1^ (A_g_), and 419 cm^–1^ (A_g_), and two low-intensity peaks at 361 cm^–1^ (B_g_) and 415 cm^–1^ (A_g_). The Raman spectroscopy obtained in the area marked by the white circle shows the A_1g_ mode corresponding to the SnS_2_ sample at 319 cm^–1^. It is worth noting that the red circle marked area shows an additional peak at 319 cm^–1^ along with the SiAs peak, which is consistent with the A_1g_ pattern of the top SnS_2_ [33,44]. The band diagram of the heterostructure in Figure 5c depicts the operating principle of the heterojunction region (2/3 electrode pair) under 550 nm laser irradiation. Since the current research on SiAs crystals is still in the realm of theoretical calculation of the bandgap of single-layer samples, the figure shows the possible positions (E_c_ ~ −3.4 eV, E_v_ ~ −6.2 eV) [22,23,24,40,41,42]. The 2D SnS_2_ crystal is a typical *n*-type semiconductor, and its band gap value has been marked in Figure 5c [44]. In order to equilibrate the Fermi level, the energy band of *p*–SiAs is inclined to *n–*SnS_2_; under light conditions, the SiAs/SnS_2_ heterojunction can absorb photons to generate the photogenerated carriers, electrons will move to SnS_2_ while holes move to SiAs, thus forming a built-in electric field. Obviously, under laser irradiation, compared with a single SiAs device, the heterostructure effectively separates the photoexcited electron-hole pairs into free charge carriers and transfers them through the interface, which helps to improve the photoelectric performance of the device.

Figure 6a shows the morphology of the SiAs–SnS_2_ *p–n* heterojunction constructed on the SiO_2_/Si substrate. Appendix A shows the results of the 4/5 electrode (SiAs device) and 1/2 electrode (SnS_2_ device). Under dark conditions, the switching ratio of SiAs devices is about 60 (Appendix A). In addition, it can be seen intuitively from the output curve that when V_DS_ = −0.9 V to +0.9 V, V_GS_ = 0 V to 80 V, it is difficult for the external gate voltage to modulate the device (Appendix A). Under 550 nm laser irradiation, with the increase in incident light power, the photocurrent changes significantly (P_in_ = 2 mW, V_GS_ = 0 V, V_DS_ = 0.9 V, I_DS_ = 44.3 pA, higher than I_Dark_ = 6.7 pA). Regarding the output curve under dark conditions, when V_GS_ = 0 V, V_DS_ = 0.9 V, and I_DS_ = 7.2 pA, we suspect that it may be the effect of oxidation on the device, and the photoelectric responsivity is close to 0.007 A/W (Appendix A). Since the photocurrent of the 1/2 electrode does not change significantly with the increase in the incident light power (Appendix A), it is difficult to obtain more accurate photoresponse data of the stand-alone SnS_2_ device. Figure 6b shows the output characteristic curves of the 2/3 electrode pair connecting the SnS_2_/SiAs heterojunction. We can see that SiAs–SnS_2_ shows obvious type II heterojunction rectification when V_DS_ = −2 V to +2 V and V_GS_ = 80 V. The forward conduction current is 100 times higher than the reverse cut-off current, showing excellent *p–n* junction characteristics.

The photoresponsivity *R* can be calculated according to its definition:(4)$$R=\frac{I_{ph}}{PA}$$
where $I_{ph}$, *P*, and *A* respectively represent the photocurrent (difference value between source–drain current and dark current under different lighting conditions), the incident optical power density (the specific information has been marked in Figure 6), and the effective irradiation area of the detector (marked by the purple curve in Figure 6a). Under light conditions, comparing the output characteristic curve of V_DS_ = −2 V to +2 V, V_GS_ = 0 V to 80 V, when V_GS_ = 0 V, V_DS_ = −1 V to +1 V, the I_DS_ changes in the heterojunction have a clear distinction, and when P_in_ = 20 μW, V_GS_ = 0 V, V_DS_ = 1 V, and I_DS_ = 0.63 nA, the result is significantly higher than the output characteristic curve of I_DS_–V_GS_ under dark conditions, and when V_GS_ = 0 V, V_DS_ = 1 V, and I_DS_ = 0.49 nA, it shows the sensitivity of the heterojunction to light conditions, as shown in Figure 6c. The relationship between the photocurrent of the SiAs–SnS_2_ heterojunction, the incident optical power density, and photoresponsivity under the condition of V_GS_ = 0 V, V_DS_ = 1 V were investigated and referred to in the results in Figure 6d,e. It can be seen that when the SiAs–SnS_2_ heterojunction is irradiated by laser at 550 nm and the power density is 0.636 mW/cm^2^, the photoresponsivity *R* is as high as 1.05 A/W (including the reasonable error range in the calculation: 0.1 A/W). With the increase in the power density, the *R* value gradually weakens and, concerning this result, we believe that although the change caused by the non-equilibrium carriers gradually increases under the illumination condition, the change in the current value is still very small relative to the increase in the power density *P*, and the result is that the *R* value gradually decreases. Our measured maximum *R* value achieves a significant improvement of nearly three orders of magnitude compared with the previously reported IVA–VA semiconductor materials of the same type, in which the maximum *R* value is close to 6 mA/W or 7.8 mA/W when the 2D SiP or 2D SiAs is irradiated by a 671 nm laser or 514.5 nm laser, respectively. [19,26] As shown in Appendix A, in order to better compare the photoresponsivity of heterojunction, we measured the absorbance of the 1/2 electrodes region (SnS_2_ structure), 2/3 electrodes region (SiAs/SnS_2_ structure), and 3/4 electrodes region (SiAs structure) in the visible range. It can be seen that the absorbance phase of SiAs/SnS_2_ heterostructure has a relative shift in the visible range, and the central wavelength is close to 550 nm. The results show that the incident wavelength of light response is reasonable. Figure 6f shows the changes in the bright and dark currents of the *p–n* junction at room temperature at V_DS_ = 1 V. It is not difficult to see that the source and drain current I_DS_ can quickly and reversibly switch between high and low states. To ensure the accuracy of the experiment and eliminate the interference of external factors, we also measured the 2/4 electrode, and the rectification ratio did not change significantly. For specific information, see Appendix A.

The detection rate $D^{*}$ can be calculated according to its definition
(5)$$D^{*}=R\sqrt{\frac{S}{2qI_{dark}}}$$
*R, S, q,* and *I_dark_* represent the responsivity, the effective irradiation area of the heterojunction, [45,46,47,48,49] the primary charge, and the dark current, respectively. Figure 7a shows specific information about the detection rate of heterostructures. It can be seen that the detection rate of the heterojunction can reach 2.7 × 10^11^ Jones when the power density is 0.636 mW/cm^2^, which is a very significant improvement compared with the single SiAs devices reported by researchers [26]. For comparison, we performed *D** calculations for the SiAs device (4/5 electrodes) at the same power density, as shown in Appendix A. *D** is close to 2.1 × 10^10^ Jones, compared with the results of a single SiAs device (4/5 electrodes), and the SiAs–SnS_2_ heterojunction has a significant performance improvement.

External quantum efficiency (EQE) is a standard evaluation index in photodetection. The external quantum efficiency is essentially the calculation of gain. It is used to study the ability of a device to collect charge and convert it into a current. It is usually an effective method for evaluating the photoelectric sensitivity of the device. The obtained value is multiplied by 100% in the calculation process, so the result is often greater than 100%. The calculation formula is:(6)$$\mathrm{EQE}=\frac{hc}{e\lambda }R_{\lambda }$$
where *h*, *c*, *e*, λ, and $R_{\lambda }$ are Planck’s constant, speed of light, actual charge, incident wavelength, and responsivity, respectively. We calculated the gain rate of SiAs/SnS_2_ under 550 nm laser irradiation. As shown in Figure 7b, it can be seen that the gain effect of the heterojunction is about 2.1 × 10^5^% when the power density is 0.636 mW/cm^2^. Compared with SiAs devices at the same power density, the heterojunction result has increased the gain effect by more than two orders of magnitude (Appendix A, EQE obtained from the 4/5 electrodes SiAs part is about 1.5 × 10^3^%), and compared with the SiAs device reported by Kim [26], it has achieved an improvement of three orders of magnitude (EQE under 514.5 nm laser irradiation is about 1.9 × 10^2^%).

### 3.6. Investigation of Photovoltaic Characteristics of p–n Junction

Due to the type II energy band characteristics of the *p–n* junction, the *p–n* junction can realize the separation of holes and electrons under light conditions, thereby forming a potential difference on the contact surface and generating a photovoltaic effect. Herein, we discuss the I_DS_–V_DS_ curve of SiAs/SnS_2_ heterojunction under laser irradiation. As shown in Figure 7c, the negative open-circuit voltage (V_OC_, Voltage at zero current) generated by the SiAs/SnS_2_ heterojunction at 550 nm, 0.52 nW laser irradiation is −0.23 V, and the positive short-circuit current (I_SC_, current at zero voltage bias) is 0.19 pA. According to the formula, electric power:(7)$$P_{el}=I_{\mathrm{DS}}\cdot V_{\mathrm{DS}}$$

Furthermore, as shown in Figure 7d, the maximum output power (*P_elMAX_*) of 17.7 fW is obtained when V_DS_ = −0.13 V. The results show that the SiAs/SnS_2_ heterojunction produces 17.7 fW of electrical power at an incident optical power of 0.52 nW and a working voltage of V = −0.13 V. Based on the equations:(8)$$FF=\frac{P_{elMax}}{V_{\mathrm{OC}}\cdot I_{\mathrm{SC}}}$$
(9)$$\eta =\frac{P_{elMax}}{P_{in}}$$

It can be calculated that the fill factor (*FF*) and power conversion efficiency (*η*) of SiAs/SnS_2_ devices are 0.43 and 0.37 × 10^−2^%, respectively. The output characteristic curve of the SiAs/SnS_2_ heterojunction shows apparent characteristics of the II heterojunction, but the small *η* value obtained by the device may be observed due to the following two reasons. First, the photovoltaic effect built-in electric field generated in the *p–n* junction mainly occurs in the space depletion area. Therefore, in our experiment, the depletion area should be very narrow, but we extracted the area of the entire device for conservative estimation. Secondly, in the process of device preparation and later evaporation, non-controllable factors such as contact between materials and air oxidation cannot be ruled out, and because the current is at the *pA* level, the error of the experiment is also challenging to ignore.

Compared with similar IVA–VA group two-dimensional layered semiconductor materials (GeAs, SiP), we conducted the first research on the *p–n* junction performance of *p–*SiAs. At the same time, the *p–n* junction constructed by *p–*SiAs has better photoelectric properties (responsivity and detection rates, etc.) than the same type of semiconductor materials previously reported.

## 4. Conclusions

The crystals are grown using the chemical vapor transport (CVT) method and then the bulk crystals are exfoliated to a few layers. Raman spectroscopic characterization has shown that the interlayer peaks in the low-wavelength band redshift increase the number of layers. *p–n* junction photoelectric devices are constructed by choosing *n*-type SnS_2_. The photoresponsivity of the SiAs–SnS_2_ heterojunction exhibits prominent rectification characteristics, and the ratio of forwarding conduction current to reverse shutdown current is close to 10^2^, showing a light response of 1 A/W under excitation light of 550 nm. The light responsivity and external quantum efficiency are increased by 100 times those of SiAs photodetectors, which is also significantly better than previous studies on such layered materials. Our work will provide experience and aid in the further construction of functional devices, such as *p–n* junctions of IVA–VA group layered semiconductors.

## Figures and Tables

**Figure 1 nanomaterials-12-00371-f001:**
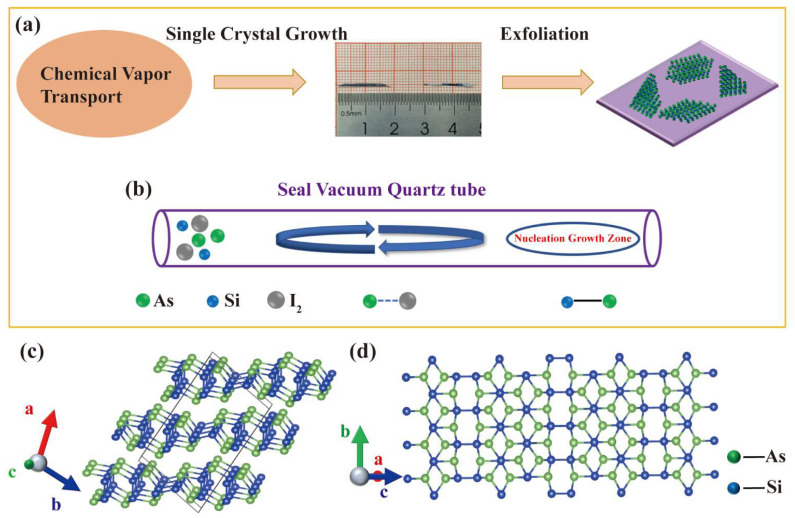
(**a**) The SiAs crystal is synthesized by the CVT method, and the 2D SiAs crystal is obtained by mechanical peeling to the target substrate. (**b**) Schematic diagram of growing SiAs single crystal in a sealed ampoule. (**c**) The monoclinic unit cell (ball-and-stick model) of layered SiAs crystal. (**d**) Top view of the atomic structure of layered SiAs crystal.

**Figure 2 nanomaterials-12-00371-f002:**
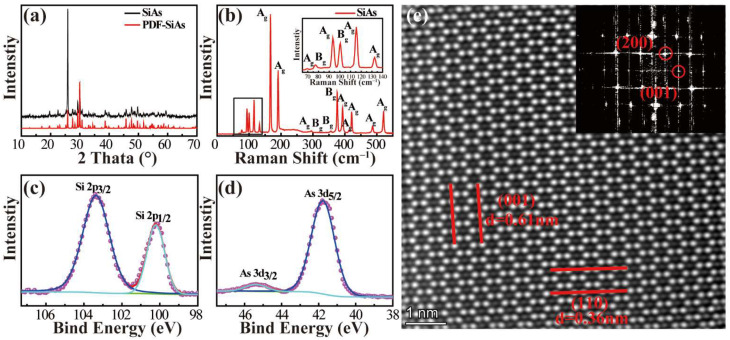
(**a**) XRD pattern of SiAs crystal powder. Take the monoclinic SiAs crystal PDF card with C2/m, a = 15.949 Å, b = 3.668 Å, c = 9.529 Å, β = 106° as a reference. (**b**) Raman spectrum of SiAs Crystals. Inset: Amplified spectrum in the range of 65 cm^–1^~140 cm^–1^. (**c**,**d**) XPS spectra of SiAs powder after grinding. Through XPS spectroscopy, we did not find the characteristic peaks of other impurity atoms, proving that the quality of the grown SiAs crystals is good. (**e**) A typical STEM image of a SiAs crystal supported on an ultra-thin carbon film.

**Figure 3 nanomaterials-12-00371-f003:**
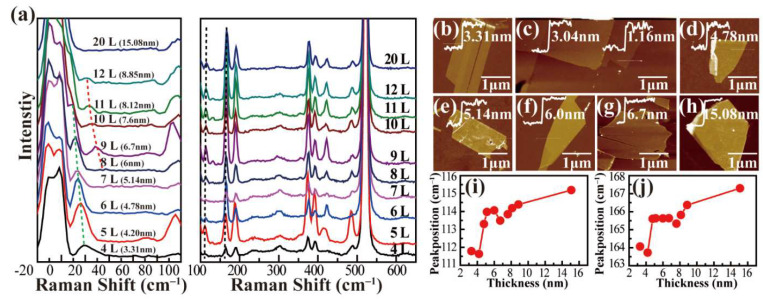
(**a**) Low wavenumber and high wavenumber Raman spectra of several layers of SiAs crystals with decreasing thickness after mechanical peeling. (**b**–**h**) AFM measurement results of some typical SiAs crystals from thin to thick in the Raman spectroscopy. (**i**,**j**) Trends in the positions of 112 cm^–1^ and 169 cm^–1^ fitted peaks of SiAs crystal with increasing thickness.

**Figure 4 nanomaterials-12-00371-f004:**
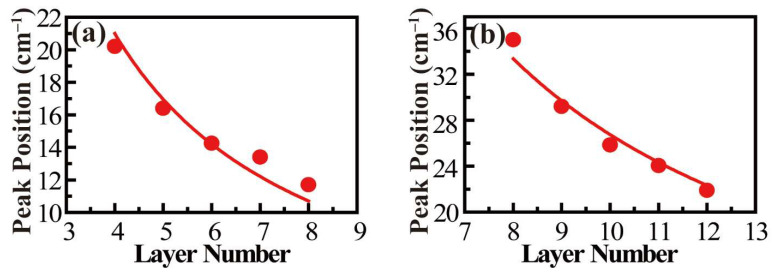
Fitting diagrams of two low wavenumber peak positions in the Raman spectrum with the obvious blue shift as the thickness decreases. (**a**,**b**) Trends in the positions of 16 cm^–1^ and 29 cm^–1^ fitted peaks of SiAs crystal with decreasing thickness.

**Figure 5 nanomaterials-12-00371-f005:**
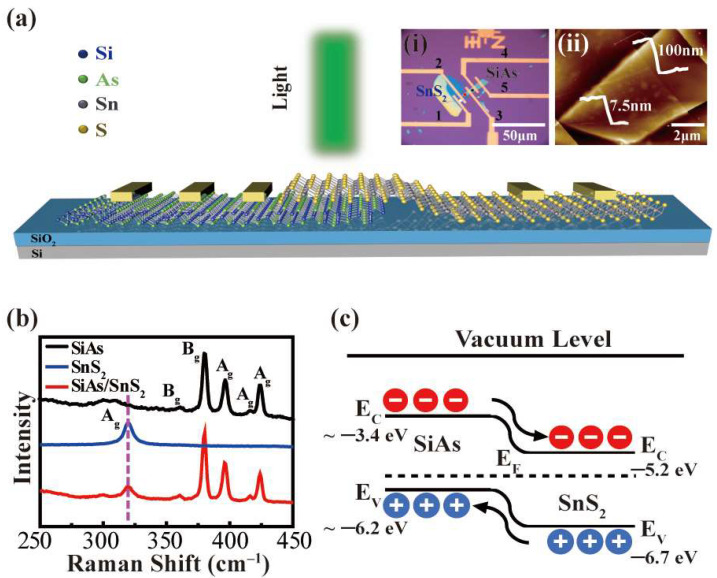
(**a**) Schematic diagram of SiAs/SnS_2_ heterojunction device configuration. Inset: (i) Optical image of a typically fabricated device. (ii) Typical AFM images of 2/3 electrode pair. (**b**) Raman characterization results at different positions in the inset of (**a**). (**c**) Schematic band bending diagram and photoexcited carriers transport the SiAs/SnS_2_ heterostructures.

**Figure 6 nanomaterials-12-00371-f006:**
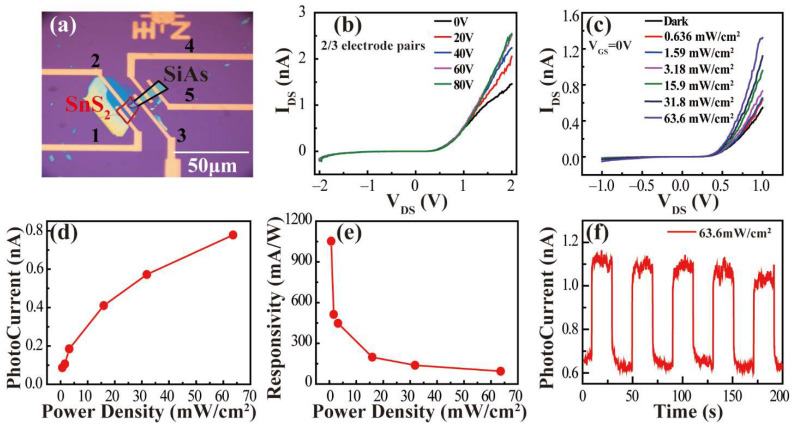
(**a**) Microscopic image of SiAs/SnS_2_ heterojunction. (**b**) The output characteristic curve of 2/3 electrode to SiAs/SnS_2_ heterojunction when V_DS_ = −2 V to 2 V. (**c**) The output characteristic curve of the 2/3 electrode pair under the conditions of V_GS_ = 0 V, V_DS_ = −1 V to 1 V, under different incident optical power. (**d**) The fitting graph of photocurrent and incident optical power density for 2/3 electrode pairs under V_GS_ = 0 V and V_DS_ = 1 V. (**e**) The relationship between detection rate and incident optical power. (**f**) Time-resolved light response when V_DS_ = 1 V.

**Figure 7 nanomaterials-12-00371-f007:**
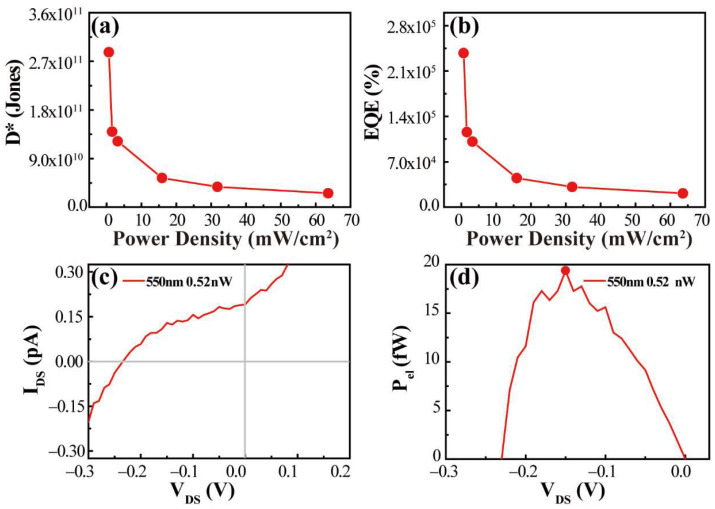
(**a**,**b**) The relationship between the *D** and *EQE* of SiAs/SnS_2_ heterojunction and the incident optical power density. (**c**) The output curve of SiAs/SnS_2_ heterojunction under 550 nm 0.52 nW laser irradiation. (**d**) Curve relationship of P_el_–V_DS_ under this laser condition.

## Data Availability

Not applicable.

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
