# Peer review of "High-Performance Photodetectors Based on the 2D SiAs/SnS2 Heterojunction"

_nanomaterials, 2022, doi:10.3390/nano12030371_

Round 1

Reviewer 1 Report

In this manuscript authors have grown the materials by chemical vapor transport (CVT) method and then exfoliated to few layers. Finally, SiAs and SnS2 have been transferred on each other to form p-n junction. The results are interesting for optoelectronic devices, but it need major revision to improve the manuscript. The points are as follow:

  1. This manuscript needs line by line a serious English editing.
  2. What about the stability of p-SiAs?
  3. There is no discussion of MoS2 in abstract and conclusion, but authors mention it in Experimental Section. Why?
  4. Raman spectroscopic characterization has shown, as the increase of the number of layers, the interlayer peaks in the low-wavelength band redshift. Why?
  5. In Figure 4a, please label that which flake belongs to SiAs and SnS2.
  6. Also, mention in Figure 4b about which electrodes are used for electrical measurements of p-n configuration.
  7. In Figure 4c, please mention that at which gate values the measurements (I-V) have been accomplished. Also, it seems that the current level under light (Figure 4c) is smaller as compared to in dark (Figure 4b). Why?
  8. In Figure 4d, please mention at which Vds value photocurrent is estimated.
  9. In Figure S3b, the output characteristics showed no variation at different gate voltages. Why?
  10. Similarly, In Figure S3d, please mention that at which gate values the measurements (I-V) have been accomplished. Also, it seems that the current level under light (Figure S3d) is smaller as compared to in dark (Figure S3b). Why?
  11. Please pay full attention to elaborate all necessary information in figures.
  12. Please draw band bending diagram of SiAs/SnS2 heterostructure.
  13. Please avoid writing Vds = -1.5V-1.5V. it confusing to readers. You can write like Vds= -1.5V to +1.5V.
  14. To make broader prospective of this manuscript, please add few recent articles of 2D heterostructured photodetectors in introduction section.  

              https://doi.org/10.1002/admi.201901304 and  https://doi.org/10.1039/D0NR05737A

  1. More important, please add the novelty and motivation of your work in the last paragraph of the introduction.
  2. Also, remove all typos throughout the manuscript.

Reviewer 2 Report

In this research paper, authors fabricate high-quality large-size SiAs single crystals by using chemical vapor transport (CVT) method and investigate the electrical and optoelectronic properties of the SiAs-SnS2 p-n heterjunction constructed on the silicon substrate.

I believe that this manuscript is well written overall and authors systematically analyzed their samples using a variety of experiment equipments (e.g. AFM, Raman). In conclulsion, I expect it will help develop next-generation electronic devices such as p-n junctions of IVA-VA group layered semiconductors. My questions I would like to mention are below.

(1) The authors mentioned a week of heat treatment time to make a bulk SiAs single crystal ("The furnace is heated to 1050 °C at a heating rate of 20 °C/min, kept for 1 week"). This fact is a major weakness of this research. Is there any way to shorten the process time?

(2) We cannot fabricate electronic devices on large areas using mechanical exfoliation methods. Can you suggest another alternative?

Thank you very much.

Reviewer 3 Report

The authors report the fabrication of a 2D based SiAs/SnS2 2 Heterojunction and the photoresponsivity to 550nm light.

A few points must be addressed:

  1. A discussion on the theoretical energy level alignment in the 2D heterojunction is missed while a scheme is present in the abstract figure. The author must discuss this point
  2. A spectral characterization of the absorbance of the whole structure must be inserted to be compared with the photoresponsivity
  3. Can the author comment on the air stability? These structures are usually encapsulated in hBN layers to prevent oxidation.
  4. It is not clear how many layers of the two materials are present in the heterostructure.
  5. What is the spot size? Does the light hit only the heterojunction or the whole region including the two materials alone?
  6. A control experiment of photoresponsivity of the two materials alone must be included

Reviewer 4 Report

This paper reports on the growth of SiAs crystals by CVT and their exfoliation. Flakes of different thicknesses are obtained and characterized using a wide set of techniques. The photoresponsivity of an heterojunction formed by exfoliated and transferred SnS2 (p type semiconductor) and SiAs (n-type) flakes on SiO2/Si is evaluated. The results are interesting and valuable but the manuscript presents several major issues that have to be fixed before being publishable. 

  1. English should be revised throughout the paper. I specify some particular points.
  2. Revise the sentence and consider splitting in two sentences: “In the past years, researchers have mostly focused on the calculation of the energy band structure of such materials and inferred that the single-layer structure has a wider band gap (about 2.5 eV, which is significantly larger than the macroscopic bulk material), the relatively wide  band gap and energy band position are conducive to the absorption of visible light photons and the  application in the field of photocatalysis”    Wider than what?, check the parenthesis.

  1. Revise the sentence and consider splitting in two sentences: “From the perspective of the intrinsic properties of the material, it is similar to the black phosphorus of the VA group, because of their high in-plane anisotropy, these materials are particularly important in the design and research of devices such as 37 photoelectric detection applications, polarization sensor devices, and angle-dependent electronic 38 devices”.   Specify which intrinsic properties?

  1. 40: “low-dimensional layer number“. This is incorrect, revise.

  1. “2D GeAs has excellent thermoelectric properties along the growth direction”   Specify which is the growth direction.

  1. “The anisotropy of 2D GeP has recently been explained more specifically”   What exactly has been explained about the anisotropy?

  1. Revise the sentence: “These research results predicted SiAs had promising applications in the future”

  1. Revise: “Until recently, Jeunghee Park et al. achieved photodetectors with single SiAs nanosheets exhibit high performance in the UV-visible region”.   What is “single SiAs nanosheets”? Perhaps the authors mean: SiAs single-layer nanosheets?

  1. Experimental: The SiAs crystals are grown using Iodine, explain the reason or/and include a reference.

  1. Include the typical sizes of the crystals in the directions perpendicular to the “strip”

  1. According to XRD the SiAs crystal grow in the (-2 0 1) direction, which does not correspond to the layers’ plane, close to (011). Comment on this and give details of the transfer process and average size of the obtained flakes.

  1. Include the description of the used Raman instrument.

  1. 92. MoS2 flakes are indicated to be used to fabricate the heterostructures but these do not appear anywhere else in the manuscript.

  1. 101: Change CTV with CVT

  1. Figure 2b: include labels for x-axis inset. Check superscripts for cm-1

  1. 128: “Among them, the Ag vibration mode is dominant. For example, the Ag vibration mode has moderate intensity peaks at…“   The use of “the Ag vibration mode” is not correct here. It should be replaced with: phonon modes with Ag symmetry.

  1. The number and symmetry of Raman modes for SiAs expected from group theory analysis should be included.

  1. 142: “The fitted value of the spectrum is close to 1.45 eV, which is roughly consistent with the previous report (calculated by PBE theory, the band gap of single-layer SiAs is around 1.7 eV).[6,7,9]."   Rephrase, it is unclear which reference applies to “the previous report” and which to “single-layer SiAs”.

  1. What is the substrate where the SiAs flakes are transferred to for Raman and AFM characterization?

  1. Experimental results on Raman phonons of few-layer SiAs are very scarce so I recommend to include Figure S2 (from Supp Info file) in the main text. This change in phonon frequencies can be very useful for other authors since measuring Raman spectra under 100 cm-1 is not common. A zoom of Raman spectra in the 100-200 cm-1 frequency range for different thickness as well for SiAs bulk crystal evidencing the frequency variations has to be included.

  1. Figure S2: “Fitting graph of the changing trend of … “. This is confusing. The authors probably mean: Trends of the positions of 112 cm-1 and 169 cm-1 fitted peaks of….

  1. (3), I suppose that N is the number of layers but it has not been defined.

  1. It is indicated that “the observed two Raman peaks can be assigned to the layer breathing modes“. How many interlayer breathing modes with clearly different frequencies (such as those here observed) are expected according to group theory analysis and calculations?

  1. How the AFM measured thickness is transformed into number of layers is unclear. Explain in detail and include in Figures 3b to 3h the associated number of layers to each of the flakes. Also, include in the spectra of Figure 3a the measured thickness of the flakes they correspond to.

  1. Figure 4: Include a schema of the heterojunction with the flakes, electrodes and gate. In (a) include the scale and identify the SnS2 and SiAs flakes. In (c) indicate the gate voltage and light power density instead of power. (d-f): indicate the used voltages.

  1. 211: “where 1/2 of the electrode increases with the incident light power, and the photocurrent does not change significantly. This is unclear, please rephrase.

  1. R in eq. (4) is not defined.

  1. l- 217: “Among them…“ Among what?

  1. 218: “For the research on the relationship between the photocurrent of the SiAs-SnS2 heterojunction and the incident optical power density and photo responsivity, please refer to the results in Figure 4(d-e).”  The reader has to interpret the figures? 

  1. Indicate the diameter of the 550 nm laser spot on the sample (perhaps in Experimental section).

  1. 229: “It can be seen that the optical responsivity R of SiAs-SnS2 heterojunction under 550 nm laser irradiation is as high as 1 A/W “. This corresponds to extremely low light power density so any error in the measured quantities produce huge errors (especially in the power density since R ~ 1/P). Specify for which power density is the given responsivity. Estimate and include the errors in responsivity in Figure 4e. 

  1. The responsivity in Figure 4e shows a drastic reduction as light power is increased. Comment on this and the implication for applications.

  1. 221: “…than the previous report of the same type of IVA-VA semiconductor materials” Include the references of “the previous reports” and the reported responsivities.

  1. 223 “In addition, Detect the photoconductivity characteristics of the p-n junction using a 550nm laser at room temperature.” Revise and rephrase

  1. The values for the detection rate and EQE are given for the lowest light power but, as photo responsivity, they drastically decrease for increasing power. These values are thus not significant and, in any case, the light power value has to be included. In order to compare the other publications this has to be taken into account.

  1. 247 “and compared with the SiAs device reported by the researchers” change to: … device reported by [Ref#].

Round 2

Reviewer 1 Report

Authors have satisfactory respond the comments. I would recommend this manuscript in present form. 

Reviewer 3 Report

The authors have addressed all the point raised in my review. Tha paper can be accepted.

Reviewer 4 Report

The manuscript can be published